# Principles and Limitations of miRNA Purification and Analysis in Whole Blood Collected during Ablation Procedure from Patients with Atrial Fibrillation

**DOI:** 10.3390/jcm13071898

**Published:** 2024-03-25

**Authors:** Mateusz Polak, Joanna Wieczorek, Malwina Botor, Aleksandra Auguścik-Duma, Andrzej Hoffmann, Anna Wnuk-Wojnar, Katarzyna Gawron, Katarzyna Mizia-Stec

**Affiliations:** 1First Department of Cardiology, School of Medicine in Katowice, Medical University of Silesia, 40-055 Katowice, Poland; 2Department of Molecular Biology and Genetics, School of Medicine in Katowice, Medical University of Silesia, 40-055 Katowice, Poland

**Keywords:** atrial fibrillation, microRNA, purification and analysis, pulmonary veins isolation

## Abstract

**Background:** MicroRNA (miRNA) have the potential to be non-invasive and attractive biomarkers for a vast number of diseases and clinical conditions; however, a reliable analysis of miRNA expression in blood samples meets a number of methodological challenges. In this report, we presented and discussed, specifically, the principles and limitations of miRNA purification and analysis in blood plasma samples collected from the left atrium during an ablation procedure on patients with atrial fibrillation (AF). **Materials and Methods:** Consecutive patients hospitalized in the First Department of Cardiology for pulmonary vein ablation were included in this study (11 with diagnosed paroxysmal AF, 14 with persistent AF, and 5 without AF hospitalized for left-sided WPW ablation—control group). Whole blood samples were collected from the left atrium after transseptal puncture during the ablation procedure of AF patients. Analysis of the set of miRNA molecules was performed in blood plasma samples using the MIHS-113ZF-12 kit and miScript microRNA PCR Array Human Cardiovascular Disease. **Results:** The miRNS concentrations were in the following ranges: paroxysmal AF: 7–23.1 ng/µL; persistent AF: 4.9–66.8 ng/µL; controls: 6.3–10.6 ng/µL. The low A260/280 ratio indicated the protein contamination and the low A260/A230 absorbance ratio suggested the contamination by hydrocarbons. Spectrophotometric measurements also indicated low concentration of nucleic acids (<10 ng/µL). Further steps of analysis revealed that the concentration of cDNA after the Real-Time PCR (using the PAXgene RNA Blood kit) reaction was higher (148.8 ng/µL vs. 68.4 ng/µL) and the obtained absorbance ratios (A260/A280 = 2.24 and A260/A230 = 2.23) indicated adequate RNA purity. **Conclusions:** Although developments in miRNA sequencing and isolation technology have improved, detection of plasma-based miRNA, low RNA content, and sequencing bias introduced during library preparation remain challenging in patients with AF. The measurement of the quantity and quality of the RNA obtained is crucial for the interpretation of an efficient RNA isolation.

## 1. Introduction

Atrial fibrillation (AF) is the most frequent arrhythmia in clinical practice, which bears a significant risk for peripheral embolism development, stroke, and death, if untreated [1]. Within the last decade, the burden of AF has almost doubled in Europe, affecting between 2% and 4% of the general population [2]. The etiology of this disease is complex, and despite of a number of clinical trials and efforts undertaken over many years, is still not completely understood. Among the crucial risk factors leading to AF are aging, arterial hypertension, heart failure, coronary disease, obesity, and diabetes mellitus [3]. Although classical risk factors demonstrated in the Framingham Heart Study [4] seem to be crucial, the genetic predisposition to AF development cannot be excluded; hence, in recent years, an increasing attention is focused on the potential ability of microRNA (miRNA) to regulate genes responsible for the onset and progression of AF [5].

The miRNA belong to the group of small, non-coding RNA molecules consisting of 19 to 25 nucleotides that bind to the untranslated regions, mainly the 3′-end regions of messenger RNA (mRNA), and regulate post-transcriptional gene expression [6]. A single miRNA can affect various genes and/or individual genes may be controlled by different miRNA, therefore creating a complex system to control gene expression during organism development, stress conditions, and pathogenesis of numerous diseases [7]. The miRNA molecules are synthesized in the nucleus, cleaved by Drosha ribonuclease III to produce pre-miRNA [8]. In the cytoplasm, pre-miRNA are cleaved by the RNase III endonuclease (Dicer) to double-stranded miRNA. The guide strand is incorporated into the RNA-induced silencing complex (RISC) and creates mature miRNA, while the other strand is degraded [9]. The miRNA and mRNA interactions are dependent on their binding complementarity and, if complete, lead to transcript degradation or inhibition of translation if they are only partially complementary to each other [10].

miRNA have potential to be non-invasive and attractive biomarkers for a vast number of diseases and clinical states. In clinical practice, miRNA can be used to identify both patients with an increased risk of AF and assess the risk of thromboembolism in the course of previously diagnosed arrhythmias. The reports from databases indicate an emergent need of searching for new biomarkers that should be able to clearly identify patients in risk groups [11]. In present studies, multiple miRNA have been proven to be involved in electrical and structural remodeling directly connected to the onset of atrial fibrillation [12]. Most of genes associated with AF encode cardiac ion channels [13] and are implicated in fibrosis and extracellular matrix structure [14], cardiogenesis [15], cell–cell coupling, and nuclear structure [16]. Dysregulation of miRNA promotes electrical disturbances and structural remodeling leading to AF. Many miRNA have been proposed as biomarkers of AF, but none currently are available for diagnostic purposes [17]. Although developments in miRNA sequencing and isolation technology have improved, detection of plasma-based miRNA, the low RNA content, and sequencing bias introduced during library preparation remain challenging [18]. In this study, we share our experiences using miRNA assays. Our aim was to show the principles of interpretation and the limitations of miRNA assays attempts. What is noteworthy in our study is how we aimed to measure and compare levels of miRNA not in peripheral blood but in blood samples from the left atrium. Whereas most miRNA studies were limited to testing miRNA levels in peripheral blood only.

## 2. Materials and Methods

### 2.1. Study Groups

This study was carried out in accordance with the Declaration of Helsinki and was approved by the Bioethics Committee of the Medical University of Silesia in Katowice (KNW/0022/KB1/9/18). Prior to this study, written informed consent was obtained from all donors. Depending on the type of AF, patients were enrolled into two study groups:Patients diagnosed with paroxysmal AF (*n* = 11);Patients with persistent AF (*n* = 14).

Both study groups were hospitalized for the ablation of the pulmonary veins. Control group (*n* = 6) comprised donors without medical history of AF who underwent ablation in the left atrium due to the presence of a left-sided accessory pathway (left-sided WPW).

The inclusion criteria for study groups (paroxysmal and persistent AF donors) included the following: aged 18–75 years; symptomatic paroxysmal or persistent AF; inefficiency of previous pharmacological antiarrhythmic treatment; and low thromboembolic risk, expressed in the CHA2DS2-VASc score: ≤1 for men and ≤2 for women.

The inclusion criteria for the control group included the following: aged 18–75 years; and no medical history of AF or other cardiovascular diseases.

The following exclusion criteria were included: diseases related to arterial pathology, neurological, and heart diseases, i.e., significant valvular heart disease, such as severe stenosis or regurgitation, thromboembolic diseases, and conditions predisposing to systemic embolism. Individuals with previous PVI or history of an interventional procedure with transseptal puncture, as well as a positive pregnancy test or renal failure with GFR < 60 mL/min, were excluded from participation. 

### 2.2. Clinical Examination of AF Patients

On admission, a physical examination was performed, and medical history was taken. A routine panel of laboratory blood tests was collected. Electrocardiography (ECG) was performed with the assessment of the leading rhythm. Standard transthoracic echocardiography was performed. Before the PVI procedure, patients from the study groups underwent transesophageal echocardiography to assess blood flow velocity and possible thrombi in the left atrial appendage.

All study groups donors (paroxysmal AF and persistent AF) were prepared for the PVI procedure with anticoagulant treatment as follows: for over 1 month before the procedure, they received non-vitamin K antagonist oral anticoagulants (NOAC)—rivaroxaban at a dose of 20 mg/day, dabigatran at a dose of 2 × 150 mg/day, or apixaban at a dose of 2 × 5 mg/day. In the groups undergoing the PVI procedure, blood was collected from the left atrium (arterial blood) after transseptal puncture. Control donors had the blood taken from the left atrium after a transseptal puncture to gain access for ablation of the left accessory tract.

### 2.3. Blood Samples Collection and Treatment

Whole blood samples from the left atrium of study and control groups were collected with the use of EDTA, mixed, incubated for 20 min at RT, and centrifuged (1900× *g*) for 10 min at 4 °C. The upper phase was transferred into a sterile tube and centrifuged (16,000× *g*) for 10 min at 4 °C. The supernatant (blood plasma) obtained was aliquoted and preserved at −80 °C for the use in the further steps of this study.

### 2.4. The miRNA Purification from Blood Samples

After the samples had been thoroughly thawed at RT, 200 µL of plasma was collected and 1 mL of QIAzol Lysis Reagent (Qiagen, Hilden, Germany) was added. Subsequently, for total RNA, i.e., containing small RNA molecules, such as miRNA, isolation procedure was conducted by the miRNeasy Serum/Plasma Kit (Qiagen, Germany) according to the manufacturer’s instructions. To optimize and compare the efficiency of the isolation method, the miRNeasy Serum/Plasma Advanced Kit (Qiagen, Germany) and the PAXgene RNA Blood kit (Qiagen, Germany) were used. All procedures were conducted according to the manufacturer’s instructions.

### 2.5. Analysis of the Concentration and Purity of RNA

The concentration and purity of collected total RNA was determined using a Nanodrop 2000 spectrophotometer (Thermo Fisher Scientific, Waltham, MA, USA) in the near ultraviolet range (200–300 nm) of light absorption, considering that nucleic acids and products of their metabolism and hydrolysis absorb ultraviolet light in the range of 250–280 nm (Table 1).

The purity of the sample was determined on the basis of the absorbance ratio at the 260 nm (typical to nucleic acids) to the absorbance of other substances, particularly proteins (280 nm) and hydrocarbons or phenol (230 nm). The ratio of A260 nm to A280 nm equal to 1.8–2.0 indicates adequate purification of the sample, whereas the values < 1.5 indicates contamination by proteins. In analogy, the A260/A230 sample absorbance ratio of 2.0–2.2 indicates pure RNA sample. The values below the value of 2.0 define the presence of hydrocarbons, including phenol, which is usually used in nucleic acids isolation procedures.

The concentration of nucleic acids was also assessed using spectrophotometric methods, considering that the absorbance of RNA at 260 nm equal to 1.0 corresponds to the concentration of 40 μg/mL. According to this rule, the miRNA concentration below 10 μg/mL indicates low efficiency of the purification procedure and contamination of nucleic acids.

### 2.6. Selective Conversion of Mature miRNA into cDNA

The total RNA template from the plasma samples was thawed on ice and then used for reverse transcription with the miScript II RT Kit (Qiagen, Germany) to generate a template for miRNA expression studies. In this method, the HiSpec Buffer was used, which allows transcription of mature miRNA into cDNA. For the reverse transcription reaction, 250 ng of template RNA was used and set up according to the manufacturer’s instructions. Subsequently, the master mix was incubated for 60 min at 37 °C and next the transcriptase was inactivated for 5 min at 95 °C and transferred on ice. The resulting cDNA was diluted in RNase-free water and used immediately for miRNA expression studies.

### 2.7. The miRNA Expression Analysis

The miRNA expression analysis was performed employing the miRNeasy Serum/Plasma Kit (Qiagen, Germany) protocol. Next, we evaluated the expression of 84 miRNA molecules, designed to monitor the development and progression of cardiovascular diseases. Initial analysis was performed using the miScript miRNA PCR Array Human Cardiovascular Disease, MIHS-113ZF-12 Kit (Qiagen, Germany), within a thermal cycler LightCycler 480 II (Roche, Penzberg, Germany). The quantitative polymerase chain reaction analysis was carried out with miScript SYBR Green PCR Kit (Qiagen, Germany) according to the manufacturer’s manual. The real-time PCR was set up at RT and 25 µL of reaction mix was placed on each well of the 96-well miScript miRNA PCR Array. The data analysis was performed according to the cycling program presented in Table 2. The Ct results were normalized to the reference genes, including the sno/snRNA miScript PCR controls (SNORD61, SNORD68, SNORD72, SNORD95, SNORD96A, and RNU6-2) according to the manufacturer’s procedure.

## 3. Results

### 3.1. Demographic and Clinical Overview of Study Donors 

The demographic and clinical characteristics of the study and control groups are depicted in Table 3. In brief, the study groups consisted of 25 patients (44% female, 56% male) at the median age of 56 years. The patients included into this study were characterized by a high symptomatology of AF (median EHRA class 3) and a low ischemic stroke risk (median CHA2DS2-VASc score of 1 pt). The patients received anticoagulation treatment with NOAC, while the routine laboratory tests, including the blood count, biochemical parameters of kidney function, as well as thyroid-stimulating hormones, were all within a normal range. We also did not observe any complications during the PVI procedure.

### 3.2. Concentration and Purity of miRNA 

Our data showed the presence of contaminated miRNA at very low total RNA yields in a standard volume (200 μL) of blood samples collected from both the studied and control groups. Low values of the A260/A230 absorption ratio are usually observed for samples contaminated with phenol, which is routinely used in nucleic acids purification procedure and/or while low concentrations of total RNA are detected. The highest concentration of miRNA was detected in patients with persistent AF; however, these samples were also the most contaminated by proteins (Table 4). Altogether, these results may indicate a prompt degradation of RNA by RNases being typical to this anatomic location (blood from the left atrium) and/or induced during the isolation procedures.

### 3.3. Optimization of miRNA Purification Procedure

Considering that low amounts of contaminated miRNA obtained in our study can result from technical issues, we performed further optimization of RNA purification.

If the concentration of purified miRNA corresponds to 10 ng/µL or below, it is recommended by the manufacturer to use TRIS-HCl buffer (10 mM, pH = 7.5) instead of RNase-free water for final elution of nucleic acids. Consequently, the miRNA purification procedure was repeated with the modification of the final stage of elution and a blank was included in the spectrophotometric measurement. As presented in Table 5, elution of tested samples with TRIS-HCl buffer resulted in decreased concentration and quality of miRNA in comparison to results observed with the use of water, indicating that this modification does not improve the miRNA purification from blood plasma samples collected from the left atrium.

For further optimization, the miRNeasy Serum/Plasma kit (Qiagen, Germany) and miRNeasy Serum/Plasma Advanced kit (Qiagen, Germany) were used to compare reagent performance under miRNA isolation procedure using two representative donors (N-14 and P-16). The miRNeasy Serum/Plasma Advanced kit (Qiagen, Germany) utilizes a modified procedure with QIAzol-free advanced chemistry without phenol/chloroform. Although, this optimization still did not allow to obtain efficient RNA isolation, the RNA concentration of tested samples was meaningfully higher using miRNeasy Serum/Plasma Advanced kit. The quantity and quality results of the purified RNA are depicted in Table 6.

The optimal quantity of blood sample needed for efficient purification of miRNA recommended by the isolation protocol manufacturer is about 200 µL. However, if the blood sample is dense, clogging of the filters by plasma components will take place, or if the levels of miRNA in blood samples are low, the final result of isolation will be inadequate. Therefore, in the next step, the isolation tests were conducted with the use of 150 µL and 400 µL of representative donor blood sample (patient P-8) by the miRNeasy Serum/Plasma Advanced kit (Qiagen, Germany) for which better performance was obtained in previous tests (Table 6). As presented in Table 7, the quantity and quality of the RNA obtained showed better efficiency of isolation procedure using smaller quantity of blood sample.

For further tests of purification procedure, the PAXgene RNA Blood kit (Qiagen, Germany) was used (patient P-8). The 42.7 ng/µL of total RNA with slight organic contamination was obtained, as confirmed by the values obtained at relevant absorption ratios, i.e., 2.1 at A260/A280 and 0.54 at A260/A230, respectively. The reverse transcription reaction was conducted using 250 ng of purified RNA. The concentration of nucleic acids (148.8 ng/µL vs. 68.4 ng/µL) as well as absorbance ratios (A260/A280 = 2.24 and A260/A230 = 2.23) indicate relatively efficient isolation and adequate purity of the RNA.

### 3.4. miRNA Expression Analysis

The obtained cDNA was used for the analysis of miRNA expression in the Real-Time PCR. The expression analysis was performed in the Roche LightCycler II device with the miScript SYBR Green PCR kit and the dedicated miScript miRNA PCR Array cardiovascular disease microarray according to the manufacturer’s protocol.

Among 84 miRNA associated with cardiovascular disease, 60 were detected (ΔCt method) in the tested, representative sample (Patient P-8), as shown at Figure 1. Among the 60 miRNA analyzed, the relative expression of 22 increased compared to the reference genes, and this increase was significantly higher for 11 of the tested miRNA (ΔCt < −2). The remaining 38 detected miRNA had lower expression compared to the reference genes, and among them 29 were significantly lower (ΔCt > 2).

## 4. Discussion

miRNA seem to be a new generation of potential biomarkers. The list of miRNA identified in human blood is increasing constantly. There are some specific miRNA identified for use in specific diseases, which require validation in larger studies [17,19]. There are several commercial kits available for isolation of microRNA from plasma. However, reports guiding the selection of appropriate kits to study are limited [20]. In our study, we measured miRNA concentration in each of the study groups. The low A260/280 and A260/A230 absorbance ratios suggested contamination. Spectrophotometric measurements also indicated low concentration of nucleic acids. Further steps of analysis revealed that the concentration of cDNA after the Real-Time PCR (using the PAXgene RNA Blood kit) reaction was higher (148.8 ng/µL vs. 68.4 ng/µL) and the obtained absorbance ratios (A260/A280 = 2.24 and A260/A230 = 2.23) indicated adequate RNA purity. The main challenge of miRNA studies includes low concentration of miRNA in blood and therefore require kits that allow optimal miRNA recovery from small volumes of serum or plasma.

miRNA have shown vital advantages as biomarkers, being implicated in regulating cellular processes, such as proliferation, differentiation, development, and cell death. They are connected to arrhythmia development processes of the cardiovascular system. In a recent meta-analysis, researchers reported miRNA-21 as negative predictor of AF that is associated with the development of myocardial fibrosis. Low expression of miRNA-150 and overexpression of miRNA-133a were also correlated with AF [21]. Other researchers showed a potential use of miRNA-21 as a biomarker of atrial fibrosis in connection with other clinical, echocardiographic, or serological markers [22]. miRNA-21 was also found to be associated with the outcome of pulmonary vein isolation. Zhou et al. found that the concentration of miR-21 in serum was strongly associated with the extent of low voltage areas in the left atrium determined by voltage mapping, which were negatively correlated with ablation success in a 1-year follow-up. Additionally, the levels of miRNA-21 in serum were associated with AF-free survival after pulmonary veins ablation [23]. miRNA can also play an important role as predictors of coronary vascular disease in patients with AF. A study conducted by Reyes-Garcia et al. showed that patients with highly increased risks of a major adverse cardiovascular event had significantly higher levels of miR-22-3p and miR-107 and lower levels of miR-146a-5p [24]. miRNA also have an important role in vascular health and atherosclerotic processes. Increased levels of miRNA-221 were shown in the serum of diabetics with microangiopathy and carotid atherosclerosis. Patients with metabolic syndrome had higher circulating miR-221 levels and patients with severe obesity had increased levels of miRNA-221 and miRNA-222 [25].

In this study, we aimed to extract and select miRNA that are connected with AF using commercial kits. We have focused on optimizing the methods of extracting and purification of miRNA.

The optimization data obtained through the assays suggest that results are affected by contaminants left after the procedure. In addition, they indicate that the efficient isolation of miRNA from tested material is impossible to achieve by using methods assumed in the research project. The low A260/A230 absorbance ratio may be due to phenol contamination or very low concentrations of nucleic acids (<10 ng/µL). The concentration and purity of isolated RNA after the PAXgene RNA Blood kit (Qiagen, Germany) indicated better results for this method compared to previous optimizations.

The main challenge of plasma miRNA studies is that their output from plasma is much lower than from solid tissues. The vast majority of miRNA function inside the cell, and miRNA that are secreted outside are in the minority. The accurate and solid measurement of miRNA in biological fluids has been difficult also due to the very short and variable nucleotide sequence [26]. This requires an appropriate preparation and preservation of samples from researcher and, in particular, at the step of material collection and nucleic acid isolation.

In our study, the plasma samples were frozen before miRNA extraction, according to protocol of isolation. The studies proved that, in terms of quantity and purity, no better results were obtained from fresh plasma than from frozen plasma samples. Also, miRNA are well preserved in frozen plasma samples so RNA can be extracted later on [27]. Many researchers suggest that plasma should be stored immediately in a freezer at −80 °C after preparation to minimize miRNA degradation. When collecting blood samples, it is advised to avoid the use of lithium heparin anticoagulation tubes because heparin can affect the process of quantification of miRNA [28].

The sample preparation is of utter importance. Blood has to be processed immediately after collection and samples should undergo two centrifugations in order to eliminate contamination by blood cells and platelets. The measurement of hemolysis is also vital, indicating that there is a possibility of interferences with specific miRNA associated with hemolysis [29]. The goal of reducing cellular content in plasma samples is important to avoid bias in miRNA, considering that in plasma samples a disease-specific miRNA concentration might be distorted by microRNAs contaminations from platelets [30]. To address this problem, in our study, we performed an extra centrifugation step.

Because of a small concentration of miRNA in circulating blood, this limitation requires kits that can allow optimal miRNA gain from small volumes of serum or plasma. Also, the sensitivity of downstream assays of miRNA is largely affected by sample contamination by residual salts from denaturing and wash buffers that are used in the extraction process. Due to the low abundance of miRNA in plasma, the isolation of particular miRNA requires suitable kit [31]. In a study by Sriram et al., the authors compared four miRNA extraction kits (miRNeasy Serum/Plasma; miRNeasy Mini Kit from Qiagen, Hilden, Germany; RNA-isolation; and Absolutely-RNA MicroRNA Kit from Agilent Technologies, Santa Clara, CA, USA). In an analysis for quality and quantity of microRNA isolation and extraction efficiency, miRNeasy Serum/Plasma kit outperformed the other three and yielded maximum microRNA quantity [32]. In our study, the highest concentration of miRNA also was achieved using miRNeasy Serum/Plasma Advanced kit by Qiagen. Measuring the quantity and quality of the RNA obtained indicates better isolation efficiency from less plasma, which may indicate that more plasma may clog the columns during the isolation procedure.

To compare the results, the PAXgene RNA Blood kit (Qiagen) was used to isolate total RNA from whole blood. In this optimization, much better results were obtained compared to previous modifications. This suggests a good quantity of miRNA and much less procedural contamination of miRNA using this kit. In recent research, we can find that, in terms of integrity, peripheral blood preserved using the PAXgene Blood RNA method had the best integrity [33].

In many scientific reports, the RNA extraction step was the main source of errors and inaccuracy in miRNA isolation process, not reverse transcription or PCR reactions. Variability of the process may be due to differences in total miRNA yield. The highest miRNA concentration was reported at a plasma volume of 200 or 300 μL. miRNA recovery level did not increase with increasing plasma volume. At plasma volumes of 400 and 500 μL, a decrease in miRNA concentration was observed [34]. We have observed similar relationship between plasma volumes in our study, where 150 µL of plasma gave better results than 400 µL. A reason for this is that when the amount of plasma is too large, it may clog the columns during the isolation procedure, increasing the loss of miRNA. Alternatively, this could have occurred because the increase plasma volume also contains higher level of contaminants which interferes with the purification process [35].

Most recommended protocols provided with the commercial kits are generic and require additional optimization to be reliably used. Optimization of protocols is required to obtain high-quality and high-yielding miRNA from small volumes of input samples in a reproducible manner [36]. Sohn at al. reported that in the absence of a reliable method of miRNA isolation from plasma, the selection of an appropriate internal or external parameter can reduce variation in results due to detection factors [37].

miRNA studies represent an attractive and promising field of investigation. Identifying and understanding the role of miRNA is an important step in the development of new therapeutic and diagnostic tools. miRNA are key molecules in arrhythmia, so their intervention and regulation has become a new target for treatment of diseases [38]. The encouraging results of miRNA applications in experimental settings and reports of negligible toxicity to healthy tissues suggest that these molecules have considerable therapeutic potential [39]. Despite the promise of miRNA-guided diagnostics, particularly in the field of minimally invasive biomarkers, several knowledge and practical issues confound or hinder translation into routine clinical practice. These include miRNA sequence database errors, suboptimal RNA extraction methods, detection assay variability, a vast array of online resources for bioinformatic analyses, and non-standardized statistical analyses for miRNA clinical testing [40].

## 5. Conclusions

miRNA have emerged as potential biomarkers, and the list of miRNA identified in human blood continues to expand. However, specific miRNA for various diseases require validation in larger studies. Selection of appropriate kits for miRNA isolation from plasma remains a challenge, with limited guidance available. Optimization of extraction and purification methods is crucial for accurate miRNA analysis. Contaminants left after the isolation procedure significantly affect the results, and efficient isolation of miRNA from plasma using standard methods is difficult to achieve. The PAXgene RNA Blood kit showed better results in terms of concentration and purity compared to previous optimizations.

The challenges of plasma miRNA studies include low output and variable nucleotide sequences. Proper sample preparation and preservation, such as immediate freezing of plasma samples, are vital for minimizing miRNA degradation. Blood samples should undergo two centrifugations to eliminate contamination by blood cells and platelets. Hemolysis measurement is crucial to identify interferences with specific miRNA associated with hemolysis. The concentration of miRNA in plasma is low, requiring kits that allow optimal miRNA recovery from small volumes of serum or plasma. Sample contamination by residual salts from extraction buffers can affect downstream assays’ sensitivity.

The miRNeasy Serum/Plasma kit by Qiagen has shown good results in terms of miRNA isolation and yield. The use of larger plasma volumes can lead to clogging of the columns and increased loss of miRNA. The RNA extraction step is often a source of errors and inaccuracy in miRNA isolation. Optimization of protocols is necessary to obtain high-quality and high-yielding miRNA in a reproducible manner.

miRNA studies hold promise for therapeutic and diagnostic applications. However, several challenges hinder their translation into routine clinical practice, including database errors, suboptimal RNA extraction methods, assay variability, bioinformatic analysis resources, and non-standardized statistical analyses for miRNA clinical testing. Despite these challenges, miRNA represent a valuable field of investigation, with potential for the development of new therapeutic and diagnostic tools. Their intervention and regulation offer new targets for the treatment of various diseases.

## Figures and Tables

**Figure 1 jcm-13-01898-f001:**
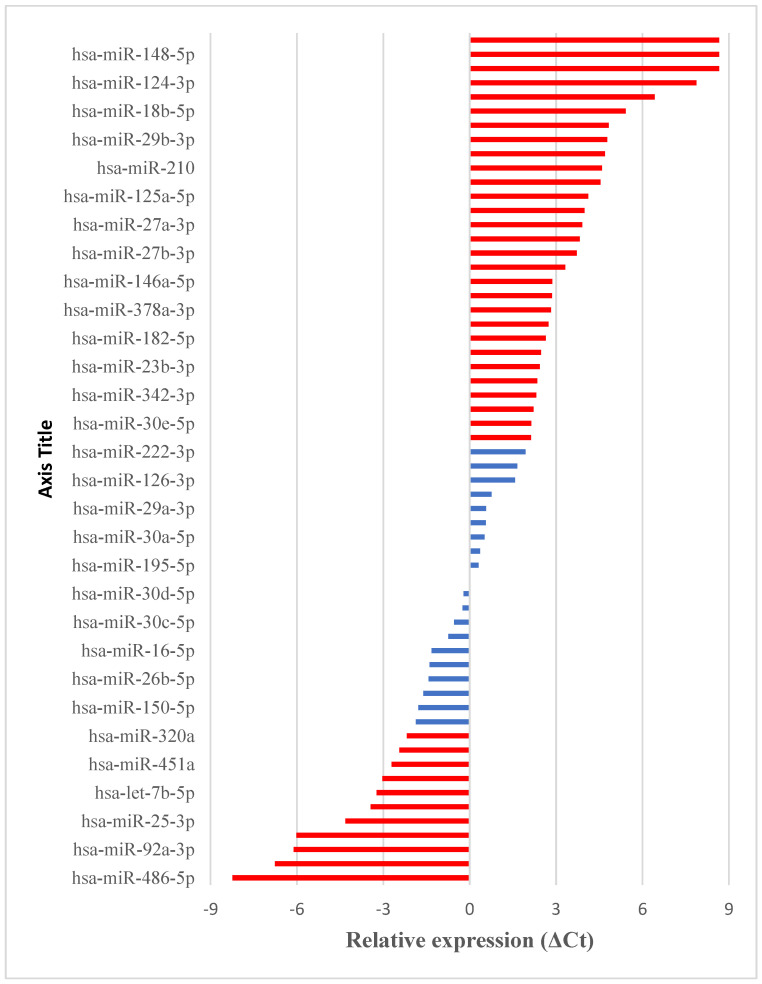
The relative expression of detected miRNA in patient from persistent AF group (P-8) blood plasma. The most dysregulated miRNA (ΔCt < −2 or ΔCt > 2) are presented in red.

**Table 1 jcm-13-01898-t001:** Maximum absorbance values for RNA and other substances present in biological samples.

Wavelength (nm)	Substance
230	EDTA, ethanol, polysaccharides
260	DNA, RNA
280	proteins
320	cellular debris

DNA—deoxyribonucleic acid; EDTA—ethylenediaminetetraacetic acid; RNA—ribonucleic acid.

**Table 2 jcm-13-01898-t002:** Cycling conditions for miRNA molecules expression analysis carried out by real-time PCR.

Step	Time [min; s]	Temperature [°C]
Initial activation step	15 min	95
**3-step cycling:**		
Denaturation	15 s	95
Annealing	30 s	55
Extension	34 s	70
**Cycle number**	**45**	**-xa**

**Table 3 jcm-13-01898-t003:** Demographic, clinical, echocardiographic, and procedural description of study groups. AF—atrial fibrillation; PVI—pulmonary veins isolation; RF—radiofrequency ablation.

Variable		Absolute Count and Percentage or Median and 25–75 Percentile or Mean ± SD
Female sex		11 (44%)
Age [years]		56 (35–66)
Body mass index [kg/m^2^]		28.44 ± 3.79
CHA2DS2-VASc [pts]		1 (1; 2)
EHRA score		3 (2b; 3)
Mode of PVI	RF	100% 0%
	Cryoablation	
AF type	Paroxysmal	44% 56%
	Persistent	
Heart failure		0
Arterial hypertension		16 (64%)
Diabetes		4 (16%)
Coronary artery disease		0
Obesity		11 (44%)
Hyperlipidemia		15 (60%)
Smoking		1 (4%)
**Pharmacotherapy:**		
Propafenone		9 (36%)
Sotalol		2 (8%)
Amiodarone		9 (36%)
**Anticoagulation:**		
Dabigatran		9 (36%)
Riwaroksaban		15 (60%)
Apiksaban		1 (4%)
**Laboratory tests:**		
Hemoglobin concentration [g/L]		14.4 ± 2.4
White blood cells [×109/L]		6.8 ± 3.8
Estimated glomerular filtration rate [mL/min/1.73 m^2^]		86.8 ± 17.8
Thyroid-stimulating hormone [uIU/mL]		1.3 (0.9; 1.8)
**Echocardiography:**		
Left ventricular ejection fraction [%]		55.4 ± 7.1
Left atrial diameter [mm]		39.2 ± 5.7

**Table 4 jcm-13-01898-t004:** Comparative analysis of total RNA concentration and purity of tested samples in study and control groups. Mean values and standard deviation (SD) are shown.

Study Design	Total RNA Concentration [ng/µL]	Absorbance Ratio A260/A280	Absorbance Ratio A260/A230
**Control (*n* = 6)**	17.27 (7.30)	1.42	0.27
**Paroxysmal AF (*n* = 14)**	17.75 (7.11)	1.31	0.28
**Persistent AF group (*n* = 6)**	25.79 (19.50)	0.64	0.43

**Table 5 jcm-13-01898-t005:** Comparison of total RNA concentration and purity results by modifying the nucleic acid elution step using RNase-free water and Tris-HCl. C—control group; N—paroxysmal AF group; P—persistent AF group; RNase-free water—ribonuclease-free water; Tris-HCL—Tris(hydroxymethyl)aminomethane hydrochloride.

Patient	Concentration of Total RNA [ng/µL]	Absorbance Ratio A260/A280	Absorbance Ratio A260/A230	Elution Step
**C-6**	15.7	0.4	0.24	**RNase-free water**
**N-1**	15.3	0.38	0.26	
**N-2**	25.8	0.65	0.42	
**N-3**	25.8	0.65	0.42	
**N-4**	12.4	0.3	0.2	
**P-4**	16.7	0.42	0.26	
**C-6**	8	0.2	0.12	**Tris-HCl**
**N-1**	9.2	0.23	0.15	
**N-2**	9.7	0.24	0.15	
**N-3**	26.5	0.66	0.5	
**N-4**	11.4	0.29	0.2	
**P-4**	6.4	0.16	0.11	

**Table 6 jcm-13-01898-t006:** Results of isolation of total RNA (including miRNA) with two purification package kits to compare reagent performance. N—paroxysmal AF group; P—persistent AF group.

Total RNA/miRNA Purification Package Kit	AF Donor	Total RNA Concentration (ng/µL)	Absorbance A260/280 Ratio	Absorbance A260/230 Ratio
miRNeasy Serum/Plasma Advanced kit	N-18	17.6	1.36	0.39
	P-7	35.8	1.45	0.43
miRNeasy Serum/Plasma kit	N-18	3.4	1.43	0.09
	P-7	4.3	1.44	0.12

**Table 7 jcm-13-01898-t007:** Comparative analysis of total RNA isolation tests with the use of 150 µL and 400 µL of representative blood sample (patient P-8) by the miRNeasy Serum/Plasma Advanced kit (Qiagen, Germany).

Amount of Plasma [µL]	Total RNA Concentration (ng/µL)	Absorbance Ratio A260/280	Absorbance Ratio A260/230
150	68.4	1.38	0.59
400	11.0	1.52	0.42

## Data Availability

Data available in a publicly accessible repository.

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
