# Peer review of "Principles and Limitations of miRNA Purification and Analysis in Whole Blood Collected during Ablation Procedure from Patients with Atrial Fibrillation"

_jcm, 2024, doi:10.3390/jcm13071898_

Round 1

Reviewer 1 Report (Previous Reviewer 1)

Comments and Suggestions for Authors

The authors are commended for making substantial improvements to the work. A few minor comments:

Line 62:  inhibition of translation, if the binding complementarity is fragmentary = or inhibition of translation if they are only partially complementary to each other.

Line 253: the word “matrice” seems unnecessary in matrice-RNA

Line 261: the paragraph break between lines 261 and 262 is not necessary and the text could flow better as “Among the 60 miRNAs analyzed, relative expression of 22 increased compared to the reference genes, and this increase was significantly higher for 11 of the tested miRNAs”.

Figure 1: perhaps colour code the data so that the 11 and 29 significantly dysregulated miRNAs are immediately distinguishable from the rest of the miRNAs.

Comments on the Quality of English Language

A couple of minor suggestions have been made in my comments above.

Author Response

Thank you very much for your revision. We have done our best to answer your comments.
We hope the changes introduced to the article as well as the answer to the revision will meet
your expectations. 
The changes introduced to the text are highlighted in red.

Comment 1:
Line 62: inhibition of translation, if the binding complementarity is fragmentary = or
inhibition of translation if they are only partially complementary to each other.

Response 1:
Thank you for the suggestions - we have corrected mentioned sentence

Comment 2:
Line 253: the word “matrice” seems unnecessary in matrice-RNA

Response 2:
Thank you for the suggestions, we have made necessary changes

Comment 3:
Line 261: the paragraph break between lines 261 and 262 is not necessary and the text could
flow better as “Among the 60 miRNAs analyzed, relative expression of 22 increased
compared to the reference genes, and this increase was significantly higher for 11 of the tested
miRNAs”

Response 3:
Thank you for your insight. We have corrected the paragraph

Comment 4:
Figure 1: perhaps colour code the data so that the 11 and 29 significantly dysregulated
miRNAs are immediately distinguishable from the rest of the miRNAs

Response 4:
Thank you for the suggestions – significantly dysregulated miRNAs are now distinguishable
from the rest of the miRNAs

Reviewer 2 Report (Previous Reviewer 2)

Comments and Suggestions for Authors

All comments have been addressed.

Author Response

Comment:

All comments have been addressed

Response:

Thank you very much for your revision. We wish to express our appreciation for the thoughtful peer-reviews that were provided by the manuscript’s Reviewers and Editor. We are glad that the article is considerably improved and has met your expectations.

Reviewer 3 Report (New Reviewer)

Comments and Suggestions for Authors

miRNA studies hold promise for therapeutic and diagnostic applications. However, several challenges, including database errors, suboptimal RNA extraction methods, assay variability, bioinformatics analysis resources and non-standardized statistical analyses for miRNA clinical testing, hinder their translation into routine clinical practice. Despite these challenges, miRNAs represent a valuable area of research with potential for the development of novel therapeutic and diagnostic tools. There are a few points to be made:

1- Include the main findings of the study in the first paragraph of the Discussion.

2-The discussion should be enriched with a comparison to the literature.

3- In addition to cardiovascular diseases in the discussion section, "Is the microRNA-221/222 Cluster Ushering in a New Age of Cardiovascular Diseases? Cor Vasa 2023;65:65-67." cite this study.

Author Response

Response to Reviewer 3 Comments

Thank you very much for your revision. We have done our best to answer your comments. We hope the changes introduced to the article as well as the answer to the revision will meet your expectations. 

The changes introduced to the text are highlighted in red.

Comment 1:

Include the main findings of the study in the first paragraph of the Discussion

Response 1:

Thank you for the suggestions - we have made necessary changes in the manuscript.

Comment 2:

The discussion should be enriched with a comparison to the literature

Response 2:

Thank you for the suggestions - we have made necessary changes in the manuscript. We have re-designed the discussion in order to meet the recommendations provided. We believe that the manuscript has been considerably improved following the peer review.

Comment 3:

In addition to cardiovascular diseases in the discussion section, "Is the microRNA-221/222 Cluster Ushering in a New Age of Cardiovascular Diseases? Cor Vasa 2023;65:65-67." cite this study.

Response 3:

We appreciate your insight. The information contained in the mentioned article has enriched the content of our manuscript.

Round 2

Reviewer 3 Report (New Reviewer)

Comments and Suggestions for Authors

I congratulate the authors for their successful editing.

This manuscript is a resubmission of an earlier submission. The following is a list of the peer review reports and author responses from that submission.

Round 1

Reviewer 1 Report

Comments and Suggestions for Authors

The authors present an interesting study that has potential to improve the evaluation and use of miRNAs in relation to atrial fibrillation.

Comments

·       From 3.2: To this goal, expression of 84 microRNA molecules with a documented association to the cardiovascular system disorders was analyzed in the whole blood samples collected from left atrium of AF donors and healthy controls”. This sentence should be removed as what follows in 3.2 is not directly related to miRNA detection.

·       Likewise Our data showed the presence of contaminated miRNA at very low concentration in standard volume (200 μl) of blood samples collected from both studied and control groups” should be re-written to say something like Our data showed the presence of contaminated miRNA at very low total RNA yields in standard volume (200 μl) of blood samples collected from both studied and control groups since the tables refer to total RNA (implying that the kits recover total RNA from the plasma, not just miRNA)

·       The N numbers are not consistent between the text in 2.1 of Methods (paroxysmal = 11 and persistent = 14) and Table 4 (paroxysmal = 16; persistent = 6). The reasons for this discrepancy need to be clearer.  

·       Were the 6 samples of Table 5 chosen at random? Why so many paroxysmal compared to control or persistent?  

·       A t-test on the datasets (RNA-free vs Tris-HCl elution) would be appropriate to confirm that the apparent difference is statistically significant.

·       Table 7: can the values obtained from 200 ul of P8 be included for comparison?

·       How much blood was used for the PAXgene RNA Blood study?

·       Do we know how much RNA should be expected from 200 µl (based on the literature or the manufacturers’ protocols)?

·       Not sure what Table 8 and the PAX gene cDNA studies add. If this were a comparison of RNA input-to-cDNA generated ratios using the different kits, that would be a different matter. In any case, what does spectrophotometric analysis of cDNA reactions reveal, considering the excess of dNTPs will also contribute to the absorbance values? Hence the last line of the abstract (Concentration of cDNA after the Real-Time PCR reaction seems to indicate much better results of this method compared to miRNA sequencing previous optimizations) is questionable.

·       Suggest re-order the graph so the data goes in order from lowest/downregulated to highest/upregulated miRNAs

·       The Discussion could do with outlining whether the differential miRNA expression seen is consistent with what has been reported in other studies.

Comments on the Quality of English Language

Quality of English Language is well.

Reviewer 2 Report

Comments and Suggestions for Authors

This is an interesting technical report on the extraction and characterisation of microRNA species from human blood of patients with atrial fibrillation. Blood has been samples both peripherally and inside the left atrium. The paper establishes a benchmark and recommends using specific commercial  kits to yield better results. A PCR array was used instead of sequencing approaches. My major recommendation is that this is a technical report, and it should be calibered as such, down-tuning the medical question. It could be AF or another disease, the principle is the same. The sample size is too small to infer any conclusion comparing microRNAs from the peripheral blood with the one inside the left atrium.

More recent and pertinent literature on microRNA and AF should be cited. For instance: https://www.ncbi.nlm.nih.gov/pmc/articles/PMC9967089/; https://www.frontiersin.org/articles/10.3389/fcvm.2023.1135127/full; https://www.ahajournals.org/doi/10.1161/CIRCEP.118.006242; https://apm.amegroups.org/article/view/86699/html.

Other previous methodological papers showing extraction of miRNA from blood have been reported: https://www.jove.com/v/20246/mirna-extraction-a-method-to-extract-mirna-from-plasma-sample; https://www.sciencedirect.com/science/article/pii/S2215016118301092; https://bmcresnotes.biomedcentral.com/articles/10.1186/s13104-019-4087-5 which could be commented on. A Discussion comparing authors method with other methods reported would be warranted.

Reviewer 3 Report

Comments and Suggestions for Authors

The topic of study would be interesting if this methodology was not being used in recent years by thousands of scientists. I do not understand the objective of the study nor do I see any sense in publishing an article of these characteristics. I don't find anything new in it or even compare microRNA amplification results in different populations. There are more reliable methods to quantify concentrations other than NANoDrop. I also do not understand what clinical data contribute to a supposedly methodological paper.